genomics; immune; personalized medicine; disease risk; non-Caucasian

**Corresponding author:**
Julian C. Knight;
Email: julian.knight@well.ox.ac.uk

# Cross-population applications of genomics to understand the risk of multifactorial traits involving inflammation and immunity

Bana Alamad[1] , Kate Elliott[1] and Julian C. Knight[1,2]

[1]Wellcome Centre for Human Genetics, Nuffield Department of Medicine, University of Oxford, Oxford, UK and [2]Chinese Academy of Medical Science Oxford Institute, Nuffield Department of Medicine, University of Oxford, Oxford, UK

## Abstract

The interplay between genetic and environmental factors plays a significant role in inter-individual variation in immune and inflammatory responses. The availability of high-throughput low-cost genotyping and next-generation sequencing has revolutionized our ability to identify human genetic variation and understand how this varies within and between populations, and the relationship with disease. In this review, we explore the potential of genomics for patient benefit, specifically in the diagnosis, prognosis and treatment of inflammatory and immune-related diseases. We summarize the knowledge arising from genetic and functional genomic approaches, and the opportunity for personalized medicine. The review covers applications in infectious diseases, rare immunodeficiencies and autoimmune diseases, illustrating advances in diagnosis and understanding risk including use of polygenic risk scores. We further explore the application for patient stratification and drug target prioritization. The review highlights a key challenge to the field arising from the lack of sufficient representation of genetically diverse populations in genomic studies. This currently limits the clinical utility of genetic-based diagnostic and risk-based applications in non-Caucasian populations. We highlight current genome projects, initiatives and biobanks from diverse populations and how this is being used to improve healthcare globally by improving our understanding of genetic susceptibility to diseases and regional pathogens such as malaria and tuberculosis. Future directions and opportunities for personalized medicine and wider application of genomics in health care are described, for the benefit of individual patients and populations worldwide.

## Impact statement

This review provides a comprehensive overview of advances in genomics within the context of immune-related diseases. It critically examines the interplay between genetic and environmental factors in determining the risk of susceptibility to immune diseases, and emphasizes the importance of considering genetic variation across diverse populations to enhance our understanding of disease etiology and allow for global application of personalized medicine. The review highlights the potential applications of genetics and wider functional genomic approaches in enhancing diagnosis, disease risk prediction, patient stratification and prioritizing drug targets, using a wide range of inflammatory and immune-related diseases as examples. It further addresses specific challenges associated with current genomic-based approaches for personalized medicine. It discusses the opportunity and limitations of polygenic risk scores, emphasizing the need for comprehensive research that encompasses genetic variation across different ethnicities and geographical regions. By critically evaluating current translational applications of genomics, the review identifies future priorities for utilizing genomic medicine for patient benefit and sheds light on ways to foster a more comprehensive and diverse scientific approach that can enable future clinical application to individuals from all ethnic backgrounds and geographical locations.

## Introduction

Heritable factors play a significant role in interindividual variation in immune and inflammatory responses. These contribute to susceptibility to disease, from very rare highly penetrant germline sequence variants causing monogenic primary immunodeficiencies (PIDs), to more common variants contributing to polygenic traits such as seen in autoimmunity or infectious diseases (Figure 1). Malaria and other pathogens have been a major selective pressure on human genetic architecture, with specific alleles driven to high frequency or fixation in some populations (Kwiatkowski, 2005; Kwok et al., 2021). While this may serve to protect individuals

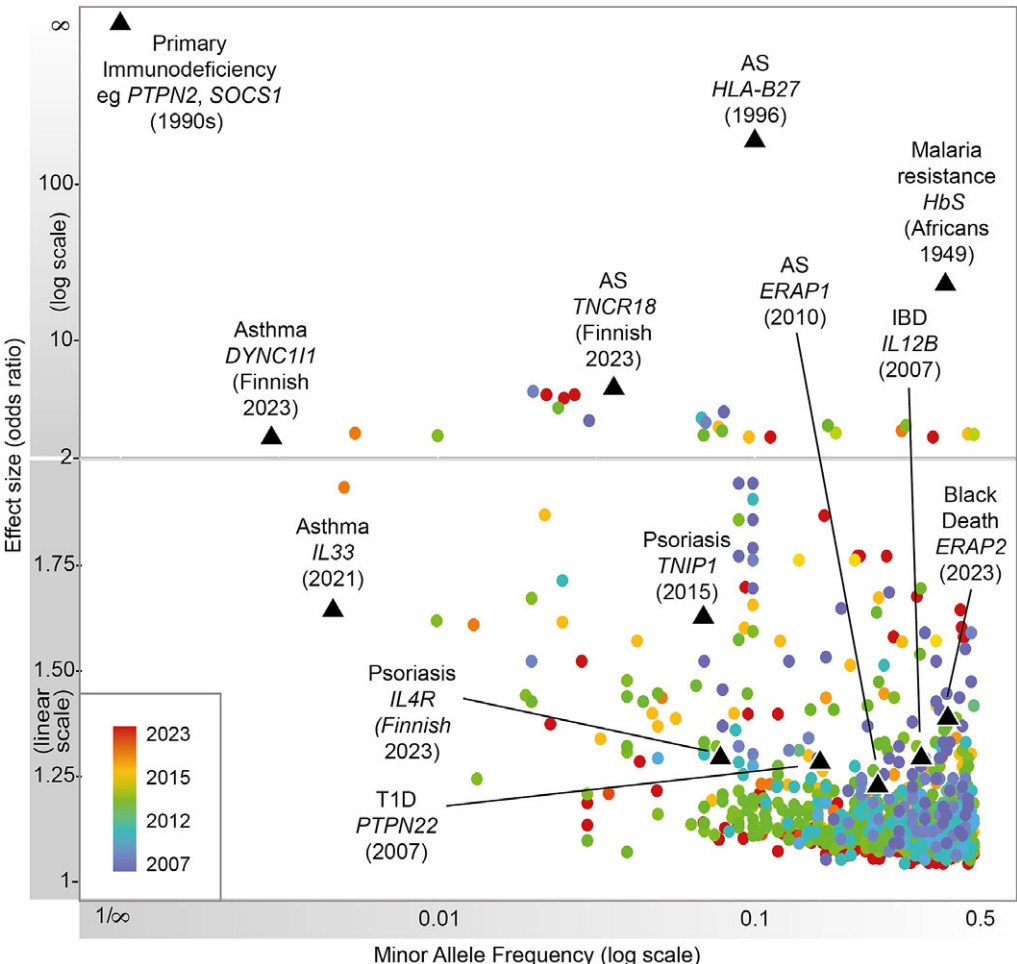

**Figure 1.** Progression of immune-related disease gene discovery over time, showing the increased power to discover variants of lower effect size with the use of larger cohorts. Black triangles refer to variants described in this review.

in high exposure environments, there may be a cost in terms of autoimmune risk such as seen with some human leukocyte antigen (HLA) alleles (Dendrou et al., 2018). The availability of high-throughput low-cost genotyping, and more recently next-generation sequencing (NGS), has revolutionized our ability to catalog such human genetic variation and understand the relationship with disease risk (Lappalainen et al., 2019). Genotyping more common single nucleotide variants has enabled genome-wide association studies (GWASs) with the identification of disease risk loci providing the opportunity for new insights into disease mechanisms, validation of drug targets and generation of polygenic risk scores (PRSs) (Sudlow et al., 2015; Tam et al., 2019; Kurki et al., 2023). NGS has significantly increased the number of known PID genes and substantially reduced the diagnostic odyssey for individual patients, while long-read technologies are allowing HLA typing at unprecedented resolution (Zhang et al., 2021; Redmond et al., 2022).

To date, our knowledge of disease genetics primarily arises from European populations (Phillips et al., 2021). High costs in establishing lab and computational infrastructure, coupled with limited expertise and training opportunities in genomics analysis, contribute to this disparity in developing countries. Budget priorities in many regions typically favor primary healthcare over research. Nevertheless, the paucity of genetic analysis in other

populations, particularly those with a higher burden of infectious disease, may lead to failure to identify risk loci and their underlying disease mechanisms, or appropriately interpret previously reported risk variants in different populations or geographical contexts, and contribute to regional inequality in the potential for genomic medicine (Figure 2). There is a global imperative to increase the representation of ethnically diverse populations in genetic studies to address knowledge disparities, reveal new disease mechanisms and ensure that genomic knowledge can be applied in an equitable and applicable way for all (Stuart et al., 2022). Mapping genetic associations in diverse populations with different genetic architectures addresses the constraints of linkage disequilibrium, which may limit the ability to fine-map specific causal alleles due to the coinheritance of variants. This allows the full spectrum of disease alleles to be defined, including where these may only be at a detectable frequency in specific populations, and enabling the identification of population-specific risk alleles, ensuring that diagnostic NGS panels and PRSs can be meaningfully applied.

Here, we will explore the potential for genomics and personalized medicine in the diagnosis, prognosis and treatment of inflammatory and immune-related diseases. We summarize the knowledge accumulated from genetic and multi-omic studies of a broad range of diseases, including immune-mediated traits and

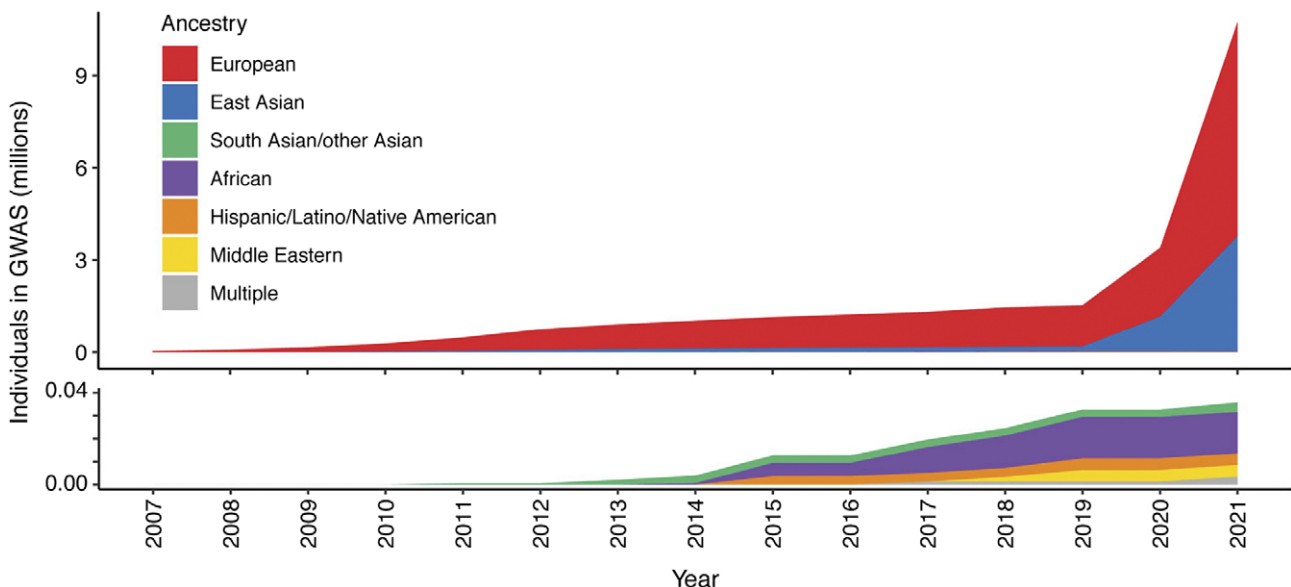

**Figure 2.** Ancestry composition of GWAS studies for 16 autoimmune traits according to GWAS catalog, per year since 2007. Disparities in the representation of global populations in GWAS show the focus on populations of European descent, with studies of other populations significantly under-represented until relatively recently. The lower panel zooms in on the y-axis scale to show the disproportionally low number of ethnically diverse GWAS studies. Figure adapted from Khunsriraksakul et al. (2022) licensed under CC BY 4.0.

infectious diseases across diverse populations, and how this is being exploited to improve healthcare globally.

## Genome variation

Large-scale population biobanks, including clinical, pathological, molecular and radiological records, have been valuable for the dissection of immune-mediated diseases, which often show overlap of clinical features and genetic pleiotropy. Globally, over 120 biobanks and population-scale datasets have now been assembled, including UK Biobank (UKB, n = 500,000) (Sudlow et al., 2015), Genomics England (n = 100,000) in the United Kingdom, Finn-Genn (n = 400,000+) in Finland (Kurki et al., 2023), deCODE genetics (n = 500,000) in Iceland (Gulcher & Stefansson, 1999), China Kadoorie Biobank (n= >510,000) (Chen et al., 2005) and Biobank Japan (n = 260,000) (Kubo, 2017). In addition to common variant association, the large size of these cohorts has allowed the identification of rarer, higher impact variants, such as the identification of an association between a rare splice in *IL33* and asthma in UKB exome data (Backman et al., 2021).

Isolated populations that experienced recent bottlenecks, like the Finnish population, provide the opportunity to identify deleterious variants, too rare to be detected in other populations. Using FinnGen data (http://www.finngen.fi/en), for 15 common diseases, 30 new disease associations were identified that were mostly low-frequency variants enriched in the Finnish population (Kurki et al., 2023). Of these, immune trait-associated variants were found near *TNCR18*, associated with inflammatory bowel disease (IBD) and ankylosing spondylitis (AS), *IL4R* associated with psoriasis (PSO) and asthma and *TNCR1* associated with PSO. This demonstrates the power of bottlenecked populations to uncover previously unknown biology of immune diseases.

Disease-specific biobanks that include tissue banking provide opportunities to generate functional genomic data at high resolution to complement genetic mapping efforts. For example, the COVID-19 autopsy biobank (Delorey et al., 2021) has been used to study the effect of severe SARS-CoV-2 infection on COVID-19 pathogenesis in different organs using multi-omic information. Here, 420 autopsy tissue samples from 11 organs were used to generate a single-cell atlas of heart, liver, kidney and lung from donors who died of COVID-19. This atlas along with lung spatial atlases presented in other studies (Melms et al., 2021; Wang et al., 2021) revealed changes in the transcriptional landscapes and cell type proportions throughout the course of infection, providing information about severe disease pathogenesis.

Capturing global population genetic variation is critical to understanding genetic susceptibility to pathogens such as malaria and tuberculosis where the burden of disease is greatest in low- and middle-income countries. Established in 2002, the International HapMap project was one of the first endeavors to catalog human genetic variation (The International HapMap Project, 2003). The aim was to map the most prevalent patterns of DNA variation (haplotypes) across the human genome by genotyping 1.3 million SNPs in 270 individuals from Africa, Europe and Asia. Subsequently, the 1,000 Genome Project (1000G), which whole-genome sequenced 2,504 individuals from worldwide populations has become a key reference resource for global human genetic variation (1000 Genomes Project Consortium, 2010). Many other population-specific genome projects have now been established including Qatar (Zayed, 2016; Mbarek et al., 2022), Saudi Arabia (Team, 2015), Iceland (Gudbjartsson et al., 2015), Uganda (Gurdasani et al., 2019), Singapore (Wu et al., 2019), Brazil (Naslavsky et al., 2022) and Japan (Okada et al., 2018).

Widely recognized as the origin of modern humans, Africans represent the most genetically diverse of human populations, as well as carrying the greatest burden of infectious disease mortality. The Human Heredity and Health in Africa (H3Africa) consortium (https://h3africa.org) (Rotimi et al., 2014) is facilitating African-led research specifically studying African genome architecture to provide insights into causal mutations for monogenic diseases and genetic and environmental risk contributions for multifactorial

diseases and traits. H3ABioNet (Mulder et al., 2016), an initiative arising from H3Africa, is addressing bioinformatics challenges in African genomics research. By promoting access and research into genetic susceptibility to prevalent infectious diseases in Africa, such initiatives are key to addressing healthcare disparities and informing global disease knowledge (Pereira et al., 2021) (Fatumo et al., 2022).

The increasing numbers of studies of common disease genetics in non-European populations has identified new loci predisposing to disease. Disease alleles of moderate or low effect size that are rare in Europeans can be detected if they exist at greater frequencies in other populations. An example is a large meta-analysis GWAS of type 1 diabetes (T1D) adding Finnish, African and East Asian samples to European samples, identifying 36 new loci (Robertson et al., 2021). In another example, in a transethnic meta-analysis, the addition of South Asian samples to European samples identified two novel psoriasis loci (Stuart et al., 2022). Inclusion of African ancestry individuals, in general, provides better opportunities for fine mapping of genetic associations as coinherited blocks of genetic variants are typically shorter with greater genetic diversity, by contrast to European haplotypes (Ge et al., 2022; Mahajan et al., 2022; Ruan et al., 2022). Such studies have the potential to identify new disease mechanisms and provide a greater understanding of disease etiology which can be used for drug development with a greater global scope.

In summary, utilizing large-scale biobanks, dedicated population genome projects and founding more regional genomics consortia remains crucial in the ongoing identification of both common and rare highly penetrant variants across diverse populations. Region-specific capacity building is therefore crucial. H3Africa and the Malaria Genomic Epidemiology Network (MalariaGEN) exemplified effective capacity building through aiding malaria-endemic countries in designing and implementing ethical research, establishing standardized methods and conducting genetic data analysis (Wonkam & Mayosi, 2014). More initiatives like this are essential to advance genomic research and impact clinical medicine.

## Infectious disease

Infectious diseases have been a strong evolutionary selective pressure on observed human genetic variation, which in some instances also modulates other immune-mediated inflammatory disease (IMID) risk. An example of this is the *ERAP2* gene where protective alleles for the Black Death caused by the bacteria *Yersinia pestis* (Klunk et al., 2022) have been found in addition to risk variants for IMIDs including AS (International Genetics of Ankylosing Spondylitis Consortium, 2013), IBD (Liu et al., 2015) and psoriasis (Tsoi et al., 2012). The Black Death is the most lethal pandemic recorded in human history and killed 75–200 million people, up to 30–50% of populations in Afro-Eurasia. Recently, susceptibility variants have been identified in DNA extracted from 516 samples from ancient burial sites in London and Denmark, genotyped at targeted immune genes. Comparison of allele frequencies before and after the Black Death identified multiple variants under selection. The most significant change was seen at an SNP, rs2549794, modulating full-length vs. truncated transcription of the *ERAP2* gene. Individuals homozygous for the protective allele were 40% more likely to survive the Black Death than those homozygous for the deleterious allele (Klunk et al., 2022). It was also demonstrated that the protective allele is associated with increased expression and production

of full-length ERAP2 protein, and macrophages harvested from individuals carrying the selected allele inhibit *Y. pestis* replication, providing a possible mechanism of resistance to the Black Death.

The sickle hemoglobin mutation (HbS), encoded by rs334, and variation in *Plasmodium falciparum* that causes malaria, also exhibit a complex balance of selective pressures between host and parasite. Homozygosity of HbS causes sickle cell anemia, whereas heterozygosity confers tolerance to *P. falciparum* infection (Kariuki & Williams, 2020; Band et al., 2022). Recently, Band et al. identified that protection against malaria conferred by HbS is dependent on parasite genotype. In an analysis of host and parasite variation in Gambian and Kenyan children with severe malaria, variation at 3 loci (Ferreira et al., 2011) in the parasite genome was found to be associated with HbS (Band et al., 2022). These HbS-associated loci include the chr2: 631,190 T > A variant within *PfACS8*, the chr2: 814,288 C > T variant within *Pf3D7_0220300*, and the chr11: 1,058,035 T > A variant within *Pf3D7_1127000*, referred to as *Pfsa1*, *Pfsa2* and *Pfsa3*, respectively. The frequency of these parasitic variants was found to correlate with HbS frequency across populations, being most frequent in Africa, where HbS is the most prevalent. This demonstrates that genetic differences between human populations can lead to different advantageous interactions with infectious parasites.

A further example involving selection is recent work which identified immune genes involved in positive and negative selection in post-Neolithic Europe, demonstrating that the historical resistance to infection and adaptation to pathogens has increased present-day inflammatory disease risk (Kerner et al., 2023). Genomic analysis has also shown how rare variants can contribute to extreme phenotypes. For example, Zhang et al. demonstrated how highly penetrant monogenic inborn errors of *TLR3* and *IRF7* are associated with severe life-threatening COVID-19 (Zhang et al., 2020). This study compared genomic sequencing data from patients diagnosed with life-threatening COVID-19 pneumonia (n = 659) and those with asymptomatic/ benign infection (n = 534). The analysis identified enrichment for rare variants in 13 gene loci involved in *TLR3*- and *IRF7*-dependent immunity, with experimental validation including evidence for a role for autosomal recessive AR deficiencies of *IRF7* and *IFNAR1* in severe COVID-19 (Zhang et al., 2020).

Genomic evidence suggests that pathogen-driven selection targeted immune-related genes contributes to inflammatory disorders (Barreiro et al., 2008; Barreiro & Quintana-Murci, 2010; Matzaraki et al., 2017; Pankratov et al., 2022). This selective pressure has led to the advantageous evolution of host defense genes, contributing to the heightened polymorphism observed in the major histocompatibility complex (MHC) (Leslie et al., 2010). In HIV-1 infection, heterozygosity at MHC class I loci confers an advantage, resulting in a slower progression to AIDS (Carrington et al., 1999). Similarly, in hepatitis B virus (HBV) infection, heterozygotes at MHC class II loci show an increased likelihood of clearing the infection (Thursz et al., 1997).

Overall, the interplay between evolutionary pressures and immune responses underscores the ongoing impact of infectious diseases on genetic landscapes shaping disease susceptibility today.

## Application of genomics for immune disease diagnosis and risk

Genomics has proven to be a powerful tool in the diagnosis of rare and complex diseases that pose diagnostic challenges. The use of

genomic sequencing for diagnosis and direct management has been used in many diseases including musculoskeletal, neurological, metabolic and complex syndromes affecting multiple tissues and processes (Ko et al., 2018; Symonds et al., 2021). Some, such as mutations in the *CFTR* gene underlying cystic fibrosis, may be tested for prenatally.

PIDs, typically resulting from highly penetrant rare mutations, are characterized by recurrent potentially life-threatening infections. To date, over 300 monogenic mutations have been identified to cause PID (GEL PanelApp) (Thaventhiran et al., 2020). Phenotypic heterogeneity in PID can make genetic diagnosis challenging (Pan-Hammarström et al., 2007; Lenardo et al., 2016; Thaventhiran et al., 2020), with only 29% of PID patients having a genetic cause of disease identified (Edgar et al., 2014). Sporadic (nonfamilial) PID presenting during adulthood can be particularly difficult to diagnose since they tend to be less severe. Suspected cases will typically undergo whole exome (WES) or genome sequencing (WGS). Noncoding variants in regulatory regions can also contribute to disease phenotypes, with, for example, colocalized novel high-penetrance monogenic variants and common variants (at the *PTPN2* and *SOCS1* loci) reported (Thaventhiran et al., 2020).

Common variable immunodeficiency disorders (CVIDs) are a heterogeneous group of PIDs. Genetic causes have been identified in 5% of CVID patients, with genotypic and phenotypic overlap between CVID and other immunological conditions reflecting the pathophysiological diversity of this disease (Peng et al., 2023). Genomic studies have shown that the majority of CVID cases are polygenic with multiple common low-penetrant variants, while a small subset of CVID patients including early onset monogenic antibody or immune deficiency, are caused by rare highly penetrant monogenic variants (Kienzler et al., 2017). *NFKB1* haploinsufficiency has been reported as the most common monogenic cause of CVID (Tuijnenburg et al., 2018), which was confirmed by Thaventhiran et al. (2020).

In terms of IMID risk, GWAS has successfully identified multiple predisposing risk variants and demonstrated a complex polygenetic architecture (Lewis & Vassos, 2020). By aggregating these genetic variants, a cumulative PRS can predict disease susceptibility and progression (Khunsriraksakul et al., 2022). PRS is calculated by summing the odds ratios of each risk allele in an individual (Lewis & Vassos, 2020; Khunsriraksakul et al., 2022) assuming an additive genetic architecture and independence between variants, which might not always be precise (Lewis & Vassos, 2020). The increasing number of studies presenting a highly significant association between PRS and disease status supports the potential for utilizing PRS as a clinical instrument, but this clinical function is still not yet robustly established.

For example, AS is a debilitating chronic inflammatory spinal arthritis affecting 20,000 people in the United Kingdom, usually presenting in young adults (Mauro et al., 2021). A hallmark of AS is the involvement of the sacroiliac joints and 85% of cases carry HLA-B27. It is one of the most heritable IMIDs (sibling relative risk ~60), of which HLA-B27 contributes ~30% of genetic risk (Khunsriraksakul et al., 2022). To date >100 loci harboring common variants have been implicated in AS risk, such as in the *ERAP1* gene (Brown et al., 2016). GWAS in AS are now sufficiently powered that PRS can be calculated that can be used to predict individual disease risk. With 78.2% positive and 100% negative predictive power, PRS has been reported to have a higher diagnostic capacity for AS than the traditional diagnostic tools, including clinical features, sacroiliac imaging, and HLA-B27 (Li et al., 2021). PRSs have also been developed for Systemic Lupus Erythematosus (SLE), a multi-organ autoimmune disease (Rönnblom & Leonard, 2019). PRS has been used to predict late-onset SLE using renal disease as a proxy for severity in both European and Chinese populations (Chen et al., 2020). Another study showed that a high genetics-based risk score is associated with an increased risk of organ damage, renal dysfunction and all-cause mortality, indicating the utility of using such scores to predict disease complications, patient deterioration, and mortality (Reid et al., 2020).

While the use of genetic information to predict risk of disease or potential drug effects can be very beneficial for personalized medicine, the lack of sufficient representation of genetically diverse populations in genomic literature limits the clinical utility of genetic-based prediction approaches like PRS in non-Caucasian populations. The predictive performance of the current PRS is lower in non-European populations. Analysis of PRS studies 2008–2017 showed that the majority of PRS studies (67%) involved European ancestry participants, with only 19% and 3.8% targeting East Asian and African/Hispanic/Indigenous populations, respectively (Duncan et al., 2019). Predictive performance was found to be lower for PRS derived from European ancestry individuals when tested in people of non-European ancestry (Duncan et al., 2019). Consistent with this, PRS derived from African American individuals greatly enhanced the predictive performance of PRS in African populations compared to European ancestry (Kamiza et al., 2022). However, the heterogeneity of African populations revealed some subpopulations performed better than others, for example, African American-derived PRS for lipid traits had a much greater predictive value in South African Zulus compared to Ugandans (Kamiza et al., 2022). The recognition of the value of data from other ethnicities is leading to an increase in data generation that will in turn improve the transferability of PRS globally.

Genetic heterogeneity within the same population can also limit PRS predictive power if the fine-scale structure within this population is not well documented at high resolution. In a Japanese cohort, phenome-wide PRS analyses on 67 complex traits were performed. Differences in PRS between the subpopulations within this Japanese cohort did not agree with the observed phenotypes for each subpopulation, suggesting that PRS differences reflect biases due to the uncorrected structure causing prediction biases in a trait-dependent manner and limiting clinical utility in non-Caucasians (Sakaue et al., 2020). Some studies also found individual-level uncertainty in PRS estimates in addition to cohort-level biases (Ding et al., 2022). A recent study by Hingorani et al. (2023) calculated the informative performance metrics for 926 PRSs across 310 diseases from the Polygenic Score Catalog. The analysis reported that PRS has a poor screening performance for individual risk prediction and population risk stratification, and indicated that the high expectations surrounding the potential of PRS for personalized medicine may be overstated.

To address the shortcomings of PRS in diverse populations, Weissbrod et al. proposed a method called PolyPred that improves PRS accuracy across populations by accounting for differences in linkage disequilibrium (Weissbrod et al., 2022). This method improved accuracy for 49 diseases/traits in some UKB populations including South Asians and Africans (Weissbrod et al., 2022). To further improve the transferability of tools like PRS, ethnically diverse genomic datasets need to be used in the development and optimization processes.

Overall, genomics stands as a robust diagnostic tool for PIDs and other rare immune- and inflammation-related diseases. While GWAS studies reveal polygenic influences on more common traits

involving such processes, the predictive potential of PRS is hindered by underrepresentation in non-Caucasian populations. Ongoing efforts are needed to generate comprehensive genomic datasets and optimized methodologies to enhance PRS clinical translation across diverse populations.

## Patient stratification and drug target prioritization

Within IMIDs, significant disease heterogeneity is recognized with the opportunity for better stratification using genetics and functional genomics. For example, multiple sclerosis (MS) is a debilitating chronic inflammatory disease affecting the central nervous system leading to irreversible neurological damage including long-term functional impairment (Pinto et al., 2020). Prediction of disease progression is currently difficult. Recent work by the International MS Genetics Consortium and MultipleMS Consortium have identified rs10191329 in the DYSF-ZNF638 locus (which includes genes involved in damage repair and control of viral infections) as associated with disease severity and specifically shortening of time to requiring a walking aid together with brain stem and cortical changes (International Multiple Sclerosis Genetics Consortium & MultipleMS Consortium, 2023).

Psoriatic arthritis (PsA) is a chronic inflammatory musculoskeletal condition that arises in about 30% of psoriasis patients (Ogdie, 2017). PsA is known to significantly reduce the patient's quality of life and increase mortality, it has various clinical presentations and often goes undiagnosed in psoriasis patients. Single-cell transcriptomics and cell-surface protein expression have been used to compare immune cell populations between PsA and PsC patients, and expression differences were successfully used as markers for psoriasis subtyping through a machine-learning model (Liu et al., 2022). Such studies demonstrate the potential utility of omics techniques in the implementation of personalized medicine.

Functional genomics can be used to stratify patients based on the nature of immune response to infection and the extent to which this is maladaptive. For example, sepsis is a highly heterogeneous clinical syndrome defined as life-threatening organ dysfunction caused by a dysregulated host response to infection (Singer et al., 2016). Transcriptomic profiling of white blood cells has revealed distinct sepsis response signatures (SRSs) that stratify a poor outcome immune-suppressed group of patients that have underlying differences in neutrophil function and emergency granulopoiesis (Davenport et al., 2016; Kwok et al., 2023). There is initial evidence that SRS is informative to define differential response to steroids (Antcliffe et al., 2019) and is applicable as a quantitative likelihood score across different infectious etiologies (Cano-Gamez et al., 2022) with the opportunity for future point-of-care testing.

Improving the efficiency of drug target selection for development is critical, given the time and cost ($2b+) of taking a drug through to approval. Attrition is high due to safety concerns or lack of efficacy (DiMasi et al., 2016), but it is recognized that genetic support increases the likelihood of success at least twofold (Nelson et al., 2015). For rare highly penetrant mutations, the modulated gene can be identified with confidence while for GWAS, typically noncoding variants are implicated, and the modulated gene may not be the nearest gene to the associated variant given the complex three-dimensional conformation of the human genome and processes of gene regulation and other mechanisms whereby a variant may exert a functional effect. The Open Targets Initiative (http://genetics.opentargets.org) integrates

GWAS data, including from UKB and FinnGen, with transcriptomic, proteomic and epigenetic data from multiple tissues and cell types using fine mapping (Mountjoy et al., 2021). Out of 133,441 published GWAS loci, this pipeline successfully identified 729 loci fine-mapped to a single-coding causal variant, and further drug target prioritization, performed by training a machine-learning model using functional validation data, improved precision and resulted in a 58% reduction in the number of false positives detected in the prioritized loci (Mountjoy et al., 2021). A study contributing data to Open Targets (Soskic et al., 2022) investigated T cell regulation during immune disease by performing single-cell transcriptomic profiling of 655,349 CD4+ T cells in healthy and activated states. Out of 6,407 genes whose expression was correlated with genetic variation between T cell states, 2,265 genes (35%) were dynamically regulated during T cell activation in immune disease providing evidence of the key genes and mechanisms underlying genetic susceptibility (Soskic et al., 2022). Other studies out of the open target initiative include single-cell genomics in asthma (Vieira Braga et al., 2019) and epigenomics identifying changes in T-cell states in immune diseases (Soskic et al., 2019).

Priority Index is a further approach to drug target prioritization for IMIDs from GWAS that provides weighting based on functional genomic evidence such as chromosomal conformation and expression quantitative trait loci in immune cells, and takes account of network connectivity based on high confidence protein–protein interactions with genetically prioritized genes to produce a predictor matrix, with prioritization scores calculated for a given gene for a particular disease where GWAS data have been inputted (Fang et al., 2019). Pathway crosstalk maximizing numbers of highly prioritized genes further identifies potential nodal genes for intervention. The ability to identify currently approved drug targets is enhanced with disease-specific functional genomic annotators, as recently demonstrated for AS (Brown et al., 2023).

Genomics plays a pivotal role in navigating the complexities of IMIDs especially through utilizing functional genomics for nuanced patient stratification. Additionally, it can support efficient drug target selection by integrating GWAS data and fine mapping. Genomics can also play an important role in vaccinology by identifying potential protective antigens and characterizing the interaction between the host, vector and pathogen (de la Fuente & Contreras, 2021). Historically, analysis of well-studied vaccines including HBV (Desombere et al., 1998), rubella (Lambert et al., 2015) and measles (Jacobson et al., 2011) vaccines have presented evidence showing genetic associations between increased antibody responses and the MHC, specifically different alleles involving HLA (Kwok et al., 2021). Genetic factors, specifically HLA type, contribute to interindividual variation in COVID-19 vaccine response and to the risk of breakthrough infection (Mentzer et al., 2023). Higher levels of antibodies against the SARS-CoV-2 spike receptor-binding domain were associated with the carriage of HLA-DQB1*06. Individuals with HLA-DQB1*06 alleles were also less likely to experience PCR-confirmed breakthrough infection (Mentzer et al., 2022). Such information may inform targeting booster vaccinations to the most vulnerable and vaccine design. Similar genetic factors may contribute to such a response in other infectious diseases, providing an opportunity to stratify individuals into groups based on their genetic information to effectively allocate vaccine distribution.

Following the identification of genetic variants, determining their pathogenicity poses challenges, especially for novel variants in disease-associated genes. To address this, computational tools

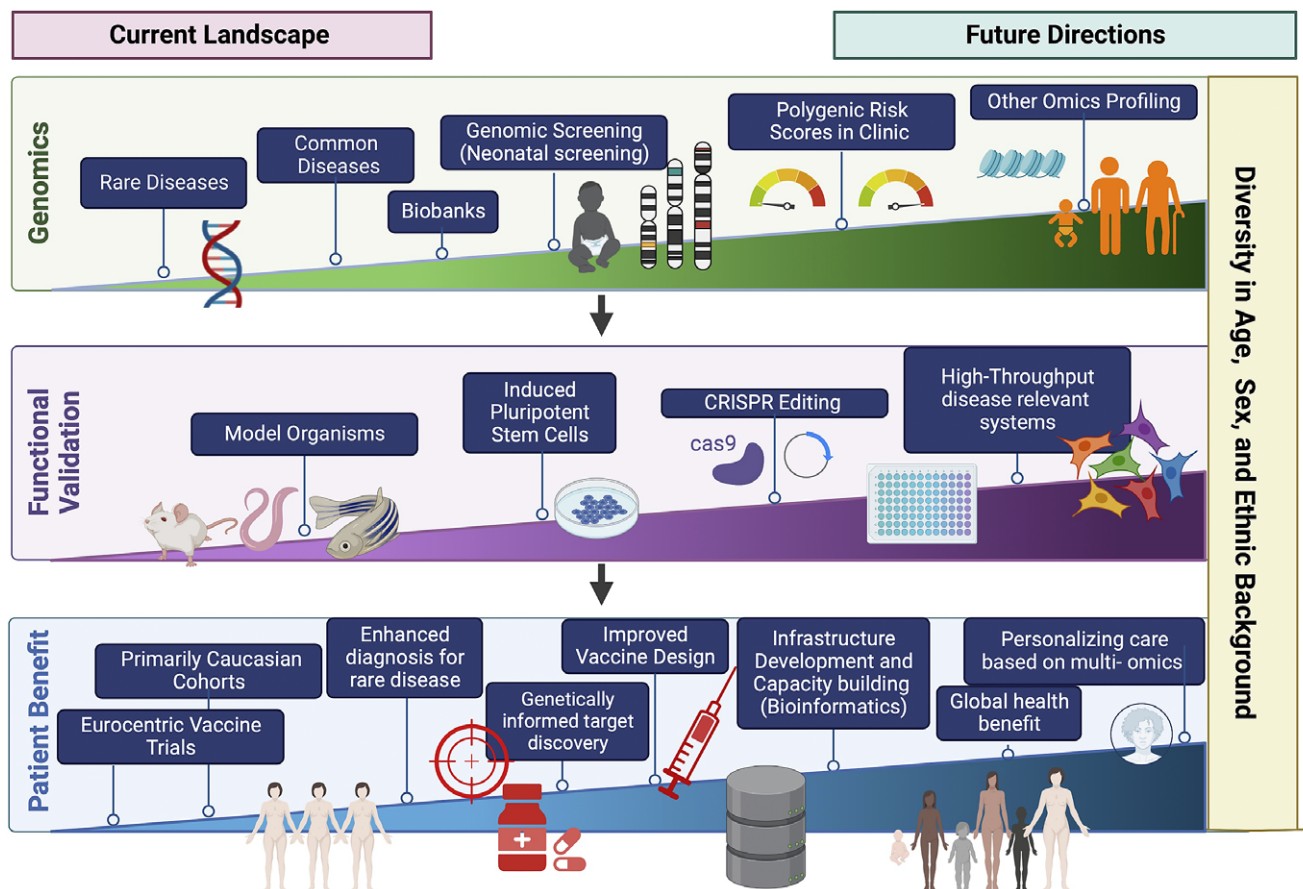

**Figure 3.** Current landscape and future directions of global genomics applications in personalized medicine highlighting key future areas in diversifying genomic medicine. Figure created with BioRender.com.

like REVEL (Ioannidis et al., 2016), AlphaMissense (Cheng et al., 2023), spliceAI (Jaganathan et al., 2019) and LoFTK (Alasiri et al., 2023) predict pathogenicity for missense, splice and loss-of-function variants while tools like DDMut (Zhou et al., 2023) predict variant effects on protein stability. With the increase in data availability and the development of machine learning and AI-based approaches, the assessment of pathogenicity for novel and known variants will improve. In addressing the challenge of clinical translation in genomics, the NHS Genomic Medicine Service, leveraging the 100,000 Genomes Project (1000 Genomes Project Consortium, 2010), stands as an exemplar to deliver genetic testing for inherited rare diseases and cancer through a National genomic test directory, supra-regional Genomic Laboratory Hubs and associated service alliances in the United Kingdom (NHS England, 2021). This infrastructure enhances the translation of genomic insights into clinical decisions for improved patient outcomes.

## Concluding remarks

Over the past few decades, advances in genotyping and sequencing technologies have allowed genomics to become an increasingly important tool in clinical medicine (Figure 3). This has expanded from the genomic diagnosis of rare diseases to the opportunity to apply PRS in predicting disease risk and outcome. However, the relative paucity of genetic studies in non-European populations limits wide applicability, including PRS. Additionally, cost and

infrastructure in many low-income countries remains a barrier to the development of genomic medicine. Advances in genomics are shaping the future of medicine, particularly contributing to the shift from reactive to proactive medicine that focuses on preventing disease rather than treating it. We need to ensure that equitable application to benefit all underpins both discovery and translational genomics with study designs involving diverse population groups worldwide, paired with appropriate knowledge sharing and capacity building.

**Open peer review.** To view the open peer review materials for this article, please visit http://doi.org/10.1017/pcm.2023.25.

**Data availability statement.** Data availability is not applicable to this article as no new data were created or analyzed in this study.

**Acknowledgments.** The authors acknowledge Dr. Calliope Dendrou, Dr. Smita Patel and Dr. Paul Wordsworth for engaging in scientific discussions about the topic of the review and providing clinical examples from their respective specialties.

**Author contribution.** B.A. wrote the first draft of the manuscript, incorporated edits and generated the final manuscript. K.E. edited and redrafted sections of the manuscript. J.C.K. edited the manuscript and oversaw the review. All authors contributed to the conception and structure of the review, read and approved the final manuscript.

**Financial support.** This work was supported by the Wellcome Trust (WT Reference Number: 204969/Z/16/Z), Chinese Academy of Medical Sciences (CAMS) Innovation 537 Fund for Medical Science (2018-I2M-2-002), the

Medical Research Council (MR/V002503/1) and the NIHR Oxford Biomedical Research Centre.

**Competing interest.** The authors declare none.

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
