## [Reviewer Report]

In general, the article was properly written, and the references cited are up-to-date. However I am not quite clear of what is the underlying message that the authors attempt to convey, except that the authors have reiterated the importance of including populations of diverse ancestries to be included in genomic study, which I could not agree more. I had difficulties in catching the major message(s) for the current flow of the storyline.

On a separate note, I was anticipating to see authors' input on the interactions between autoimmune response and infectious diseases but somehow I was not able to catch this clearly in the manuscript. In addition, infectious diseases like HIV was not elaborated, was there a reason behind? FCGR gene cluster variations (SNPs and CNVs), perhaps would be another consideration in this manuscript?

Last, whilst authors had raised several challenges in genomic study, perhaps author could improve the argument better by providing thoughts on the underlying reason(s) for these challenges, subsequently may consider providing some recommendations to address the challenges the subject matter.

Overall, I am in favour of publishing this manuscript, if the above comments could be appropriately addressed or explained.

---

## [Reviewer Report]

The manuscript focuses on the application of genomics in understanding and treating immune-related diseases. The review is informative, concise and thought-provoking. It underscores the role of genomics in enhancing diagnosis, prognosis, and treatment plans. It also covers difficulties of implementing genomic-based approaches like polygenic risk scores in clinical settings, especially for diverse populations. The authors make a good point stressing the need to consider genetic variations across different populations to better understand disease etiology. The review also outlines the limitations and future milestones in the field.

I have two minor comments which could improve this already excellent and timely review:

1. Please make sure to include which gene particular mutation is affecting (e.g. it’s not mentioned in the case of T14484C). This would add clarity to the manuscript.

2. One of the challenges in the field is making an informed decision based on DNA sequencing result, which reports a potentially pathogenic mutation in a gene usually associated with a particular disease, however the mutation seems to be novel. Some recently developed computational tools involving large language models and deep learning (e.g. AlphaMissense, DDMut) might aid in assessing the pathogenicity of such mutations. It would be good to mention these challenges and discuss the potential future developments in this area.